# Unique Dye-Sensitized Solar Cell Using Carbon Nanotube Composite Papers with Gel Electrolyte

Yi Kou * and Takahide Oya

Graduate School of Engineering Science, Yokohama National University, Yokohama 240-8501, Japan;
oya-takahide-vx@ynu.ac.jp
* Correspondence: kou-yi-zt@ynu.jp

**Abstract:** We propose a unique form of dye-sensitized solar cells (DSSCs), paper DSSCs based on carbon-nanotube (CNT) composite papers, and the use of a gel electrolyte for the paper DSSC. In our previous study, we succeeded in developing the paper DSSC. However, its performance and lifetime were not sufficient. We considered that the problem was the use of liquid-type electrolyte. To improve the performance of the paper DSSC, a gel electrolyte was introduced to increase safety and durability. Here, a polymer gel electrolyte was synthesized using a copolymer of polyethylene glycol (PEG) and polyvinylidene fluoride (PVDF) as a matrix, mixed with iodine and potassium iodide. The resulting paper DSSC had a fill factor (*FF*, a performance indicator) of 0.248 and a conversion efficiency of $2.43 \times 10^{-5}\%$ with an extended working time (lifetime) of more than 110 min. Further modifications were made to the metallic CNT composite paper and the gel electrolyte, resulting in an increased conversion efficiency of $2.02 \times 10^{-3}\%$. This study suggests the potential of gel electrolytes in enhancing the performance of paper DSSCs, providing new insights for their future applications.

**Keywords:** carbon nanotube (CNT); carbon nanotube composite paper; dye-sensitized solar cell (DSSC); paper DSSC; gel electrolyte

## 1. Introduction

Renewable energies such as solar, wind, hydro, biomass, and geothermal energy are gaining attention as a solution to the energy and environmental problems that have arisen in recent years. These renewable energies are continuously regenerated in nature and can be used indefinitely. In particular, solar power generation has been studied and used for a long time and is currently important.

One promising area of research in solar energy is the development of dye-sensitized solar cells (DSSCs), also known as Grätzel cells [1,2], which are focused on in this study. Developed DSSCs consist of a negative electrode with dye-attached semiconducting particles, electrolyte, and a positive electrode. They use a layer of dye molecules to capture light and convert it into electricity. The dye molecules are contained within a porous layer of titanium dioxide as the semiconducting particles and are surrounded by an electrolyte solution. When light is absorbed by the dye molecules, electrons are released and move through the titanium dioxide ($TiO_2$) to the electrode, generating a current. It is known that $TiO_2$ is very important in many fields [3] and has an important role for the DSSC. The power generation principle is illustrated in Figure 1 [4].

The power generation process of the DSSC can be described in the following four steps:

(1) By receiving solar light, electrons in the dye molecule jump from the ground state to the excited state.

(2) The excited-state dye molecule injects an electron into the conduction band of the semiconductor (particle), and the electron diffuses to the conductive substrate (electrode) and flows into the external circuit.

(3) The oxidized dye because of releasing excited electrons is reduced by the reducing electrolyte.

(4) The oxidized electrolyte receives electrons at the counter electrode and is reduced, completing one cycle.

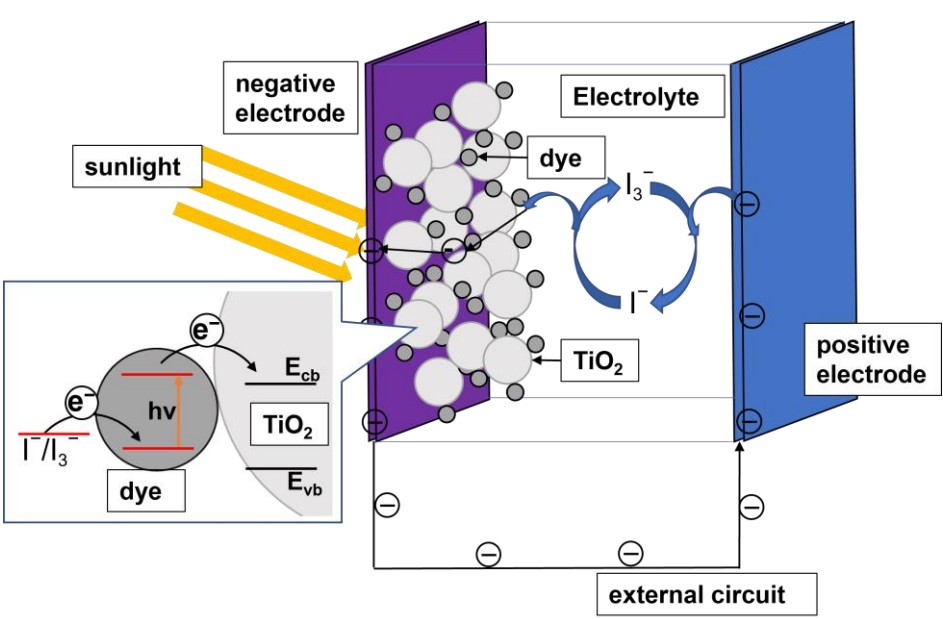

**Figure 1.** Operating principle of dye-sensitized solar cell.

Unlike traditional silicon-based solar cells, DSSCs can be produced using low-cost materials and are lightweight, flexible, and semi-transparent. These characteristics make them an attractive option for a wide range of applications, from portable electronics to building-integrated photovoltaics.

Carbon nanotubes (CNTs) were first discovered in 1991 [5]. They are classified into two main types: single-walled CNTs (SWCNTs) and multi-walled CNTs (MWCNTs). SWCNTs have a cylindrical structure in which carbon atoms are arranged in a single layer, while MWCNTs are formed by folding multiple coaxial graphene sheets. The unique properties of CNTs, such as their semiconducting or metallic properties, electrical conductivity, and high thermal conductivity, have made them a focus of research in various fields.

In 2007, we developed a unique CNT composite, a CNT composite paper [6], by combining CNTs and pulps (raw materials of a paper), as shown in Figure 2. The paper can be easily produced using a method inspired by the traditional Japanese *washi* papermaking process, and it has the potential to be used in various devices. Practically, many useful applications based on the CNT composite papers have been developed, such as a thermoelectric power generating paper [7,8], a paper actuator [9], a paper antenna [10], an electromagnetic shielding paper [11], keys for an artifact-metric authentication system [12,13], and a paper transistor [14]. The unique properties of CNTs, such as their metallic and semiconducting properties, make them suitable for use as both positive and negative electrodes, including as semiconducting particles for dyes in DSSCs. Therefore, we believe that our CNT composite paper has the potential to be a sustainable and environmentally friendly alternative to traditional DSSCs. With minimal environmental impact, this paper-based DSSC (paper DSSC) technology could pave the way for the development of low-cost and eco-friendly solar energy harvesting technology.

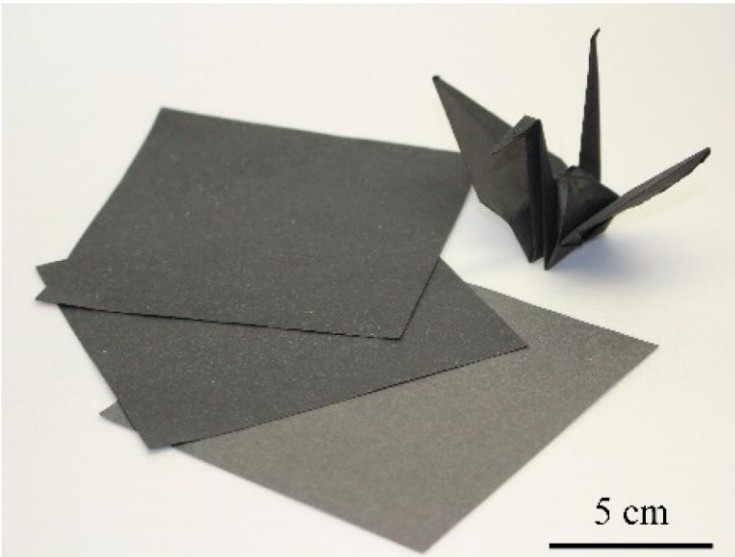

**Figure 2.** CNT composite papers. Differences of color are caused by volume of contained CNTs. (From Ref. [15] under License CC BY 4.0.)

In our previous study [15], we succeeded in developing a paper DSSC using our CNT composite paper. Then, the CNT fraction and electrolyte were subsequently modified to achieve a photovoltaic conversion efficiency of 0.18%. However, the use of liquid electrolyte caused the electrolyte to evaporate and leak after 20 min of operation, resulting in the device's failure. Therefore, this study aims to optimize the stability of our paper DSSC by introducing a gel electrolyte and further investigate the enhancement of its photoelectric conversion efficiency. This introduction of gel electrolytes is expected to suppress electrolyte evaporation and improve performance.

## 2. Materials and Methods

This study aimed to optimize the stability and efficiency of our paper DSSC by using gel electrolytes. To achieve this, three types of CNTs, namely SGCNT, HiPco, and (6,5)-chirality CNTs, were used to prepare the composite papers. Gel electrolytes were then prepared using polyethylene glycol (PEG) and poly-vinylidene fluoride (PVDF) with reference to Ref. [16] and introduced into the composite papers. The resulting gel electrolyte-introduced CNT composite papers were used to create our new paper DSSCs, and their fill factor and conversion efficiency were measured.

### 2.1. Method of Manufacturing Carbon Nanotube Composite Paper

In this study, two types of CNTs, metallic and semiconducting types, were used to manufacture the CNT composite papers to construct our paper-DSSC. The metallic CNT composite paper was used for the positive electrode, while the semiconducting CNT composite paper was used for the negative electrode in our DSSCs, respectively. Our CNT composite paper manufacturing procedure is based on the traditional Japanese *washi* manufacturing method, as described in our previous studies [6,15].

Figure 3 shows the actual manufacturing process of our composite papers. As shown in Figure 3, the manufacturing method is the same, and either metallic or semiconducting CNT composite paper can be easily produced by simply changing the CNTs used.

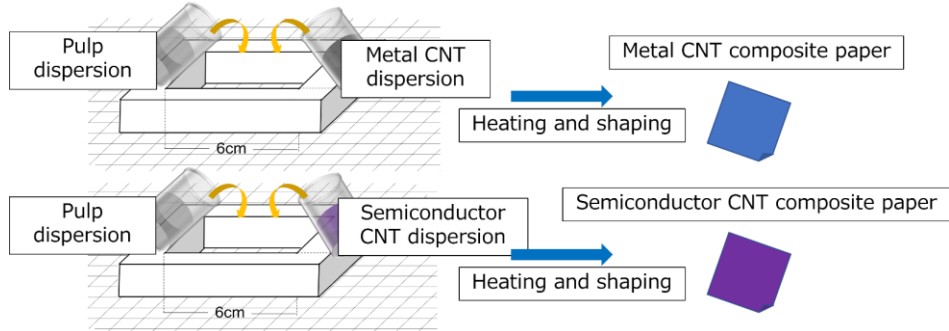

**Figure 3.** Schematic of manufacturing method for CNT composite paper.

As an example, the specific process of metallic CNT composite paper fabrication is as follows:

(1) Add 1 g of pulp (raw material of a paper) to 100 mL of pure water and mix thoroughly with a mixer until the pulp lumps are sufficiently loosened to prepare a pulp dispersion.
(2) Mix 48 mg of SWCNTs (SGCNT, SG101, ZEON CORPORATION, Tokyo, Japan) and 70 mg of sodium dodecyl sulfate (SDS) as dispersant in 36 mL of pure water.
(3) Use an ultrasonic homogenizer (UX-50, Mitsui Electric Co., Ltd., Tokyo, Japan) to irradiate the mixture from step (2) with ultrasonication for 90 min to prepare a CNT dispersion.
(4) Further disperse the CNT dispersion from step (3) by adding 45 mL of the pulp dispersion from step (1) and mixing it thoroughly.
(5) Use the method described in Figure 3 to manufacture metallic CNT composite paper from the dispersion prepared in step (4).

Specific ingredient quantities are shown in Table 1 below:

**Table 1.** Amount of materials used for metallic CNT composite paper.

| For Pulp Dispersion | | For CNT Dispersion | | |
|---|---|---|---|---|
| Pulp | Pure Water | CNT | SDS | Pure Water |
| 450 mg | 45 mL | 48 mg | 70 mg | 36 mL |

The same manufacturing method described above is used in the fabrication of semiconducting CNT composite paper. Here, this study focuses on purple sweet potato dye (anthocyanin dye, PSP-135P, Kiriya Chemical Co., Ltd., Osaka, Japan) as a dye for the DSSC. This is because it is an inexpensive, natural dye and has a good performance as a dye for the DSSC. Figure 4 shows the chemical structural formula of purple sweet potato dye. It is widely known that molecules containing benzene rings and hydroxyl groups, such as catechin [17,18], have the ability to disperse CNTs in water through π–π stacking interaction. As shown in Figure 4, this molecule also has several benzene ring structures, which we considered to have the ability to disperse CNTs, like catechins. In addition, the dye must be fixed on the semiconductor electrode for the DSSC fabrication. Therefore, purple sweet potato dye was used as a dispersing agent instead of SDS in this study. Table 2 shows the specific ingredient quantities to manufacture our semiconducting CNT composite paper. Here, we use (6,5)-chirality CNT (SG65i, CHASM) as a semiconducting CNT. The manufactured CNT composite papers are shown in Figure 5.

**Figure 4.** Chemical structural formula of purple sweet potato dye.

**Table 2.** Semiconducting CNT composite paper.

| For Pulp Dispersion | | For CNT Dispersion | | |
|---|---|---|---|---|
| **Pulp** | **Pure Water** | **CNT** | **Purple Sweet Potato Dye** | **Pure Water** |
| 360 mg | 36 mL | 4 mg | 36 mg | 18 mL |

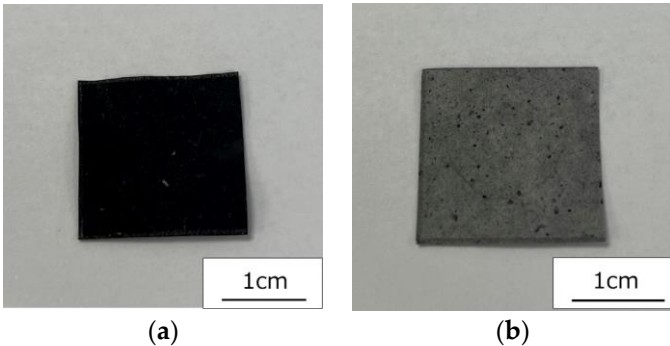

(**a**)          (**b**)

**Figure 5.** Manufactured CNT composite papers. (**a**) Metallic type and (**b**) semiconducting type.

### 2.2. Methods of Making Gel Electrolytes

In recent years, gel electrolytes have attracted attention as electrolytes for DSSC. For example, studies on the use of polyethylene oxide (PEO) [19], poly-acrylic acid-polyethylene glycol (PAA-PEG) [20], polymethyl methacrylate (PMMA) [21], and polyvinylidene fluoride-hexafluoropropylene copolymer (PVDF-HFP) [22] are being conducted. In ordinary liquid electrolytes, evaporation and leakage may cause degradation of power generation performance. On the other hand, the gel electrolytes have the potential to eliminate those problems. In this study, we introduce a new gel electrolyte as an electrolyte for our DSSC using CNT composite papers. Specifically, the use of PEG-PVDF (polyethylene glycol-polyvinylidene fluoride) gel electrolyte is considered. The specific preparation process of the electrolyte is shown in Figure 6.

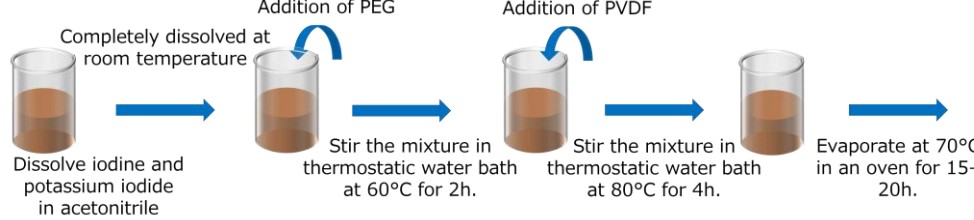

**Figure 6.** Schematic of producing method for gel electrolyte.

The specific process for preparing the PEG-PVDF gel electrolyte for our paper DSSCs is as follows:

(1)  Dissolve 19.1 mg of iodine and 100 mg of potassium iodide in 10 mL of acetonitrile by immersing the phial with the solution in a constant temperature bath at 60 °C.

(2)  After the complete dissolution of the iodine and potassium iodide, add 0.5 g of PEG under the same conditions and wait for approximately 2 h until it is completely dissolved.

(3)  After 2 h, add 0.5 g of PVDF and wait for approximately 4 h until it is completely dissolved at 80 °C to obtain the electrolyte solution.

(4)  Put the phial with this solution in an oven at 80 °C for 15 h to evaporate most of the acetonitrile and obtain a viscous electrolyte solution.

(5)  Finally, the solution begins to gel when allowed to stand at room temperature, resulting in a gel electrolyte (Figure 7). The gelation is performed on our CNT composite paper as described below.

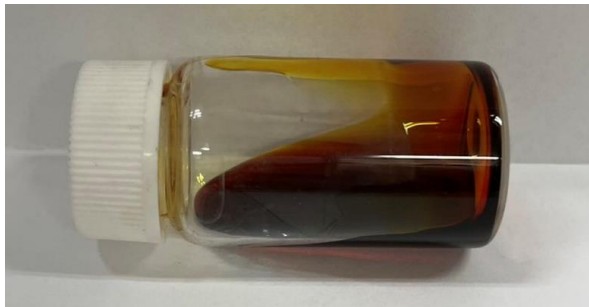

**Figure 7.** Obtained viscous gel electrolyte.

## 2.3. Assembly and Performance Evaluation Method of Paper Dye-Sensitized Solar Cells

For the fabrication of the new paper DSSC, the metallic CNT composite paper was used as the positive electrode, while the semiconducting CNT composite paper was used as the negative electrode. A total of 400 µL of the gel electrolyte obtained by following the process described in Section 2.2 was dropped onto both CNT composite papers. The electrodes were then left to allow the gel electrolyte to solidify between them. This process resulted in the creation of a new paper DSSC as shown in Figure 8. The actual samples fabricated using this method are shown in Figure 9. In the actual sample preparation, the electrolyte was introduced from the surface of the paper for fabrication convenience. As a result, the sample has a yellowish color resulting from the electrolyte, but the tint of the composite paper with dye before the introduction of the electrolyte, shown in Figure 9, has a purple color resulting from the dye.

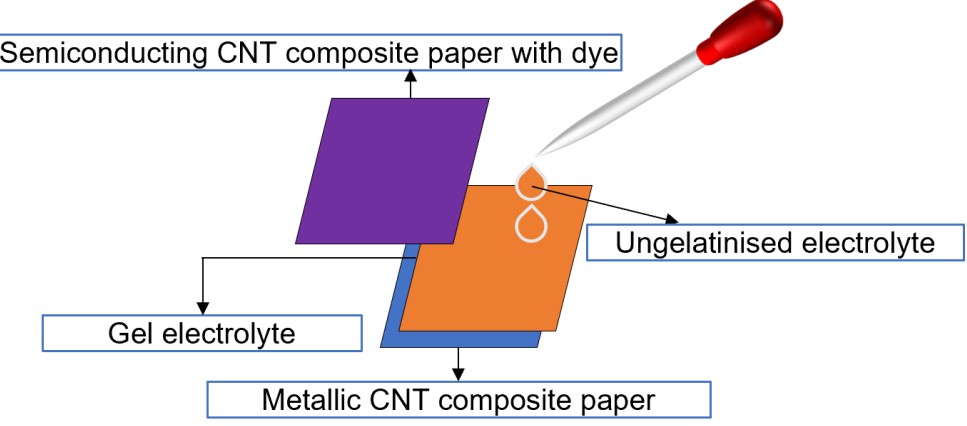

**Figure 8.** Schematic of new paper DSSC.

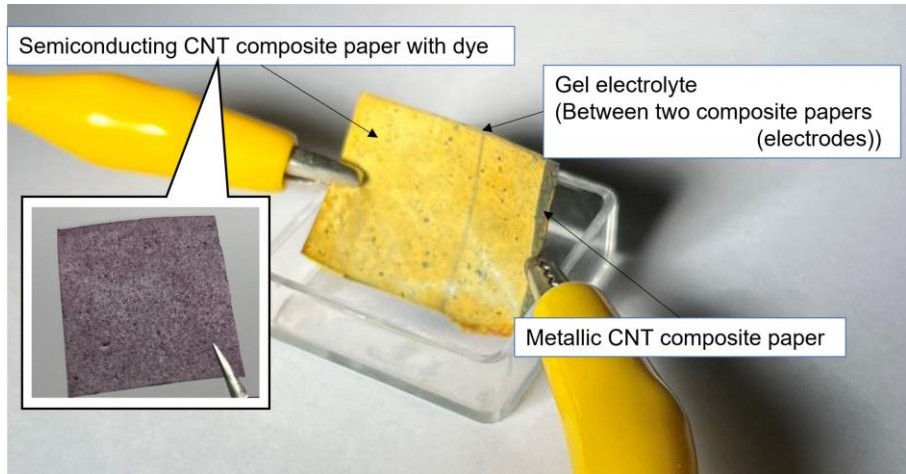

**Figure 9.** Fabricated paper DSSC.

Generally, the performance of DSSCs is primarily evaluated by measuring their short-circuit current, $I_{SC}$, and open-circuit voltage, $V_{OC}$, followed by the calculation of fill factor, *FF*, and photoelectric conversion efficiency, $\eta$. *FF* is determined by the ratio of the product of current and voltage at the maximum power $P_{MAX}$ ($= I_M \times V_M$) point to the product of $I_{SC}$ and $V_{OC}$ as shown in Figure 10. A larger hatched area in the figure indicates a higher *FF*, which results in better output performance of the DSSC. It is known that *FF* is mainly influenced by the type of electrolyte and the impedance of the cell.

$$FF = \frac{P_{MAX}}{I_{SC} \cdot V_{OC}} = \frac{I_M \cdot V_M}{I_{SC} \cdot V_{OC}} \tag{1}$$

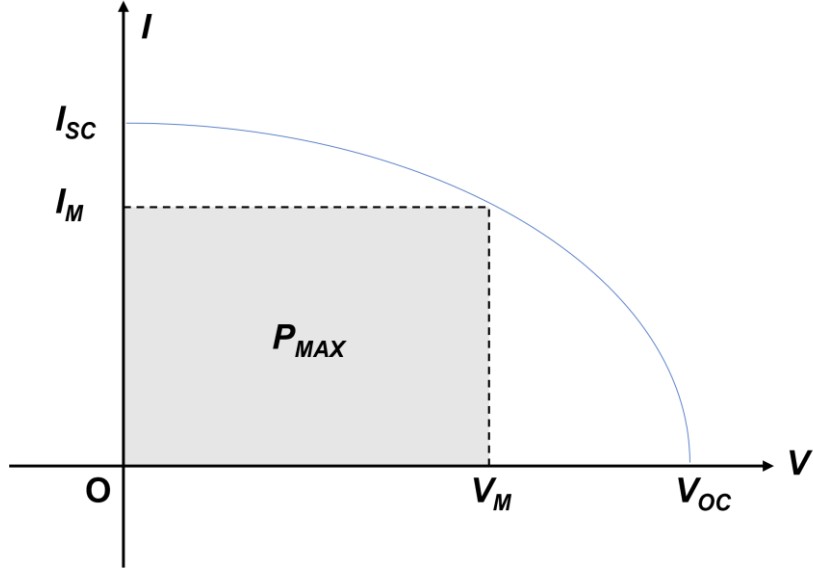

**Figure 10.** Desired *I-V* characteristics of solar cells.

$\eta$ is calculated as the ratio of $P_{MAX}$ obtained under irradiation to sunlight (or artificial light) $P_{in}$. This value is obtained by combining $I_{SC}$, $V_{OC}$, and *FF* according to the following equation.

$$\eta = \frac{P_{MAX}}{P_{in} \cdot S_{active}} = \frac{I_{SC} \cdot V_{OC} \cdot FF}{P_{in} \cdot S_{active}} = \frac{J_{SC} \cdot V_{OC} \cdot FF}{P_{in}} \tag{2}$$

where $S_{active}$ indicates the area of the region irradiated by light and $J_{SC}$ is the current density per unit area.

To investigate *I-V* characteristics of the samples prepared in this study, measurements were conducted using a semiconductor parameter analyzer (Keithley, Semiconductor Characterization System, 4200A-SCS, Solon, OH, USA). The effective area of the sample was 3 cm$^2$, and artificial sunlight with an intensity of 1000 W/m$^2$ was used for irradiation. The measurements were taken every 2 min using the semiconductor parameter analyzer over a period of 10 min.

## 3. Results and Discussion

We prepared our paper DSSCs and evaluated their performance following the method described in Section 2.3. Figure 11 shows the *I-V* characteristics of the sample when operating under artificial sunlight for a period of 10 min. Based on the calculations, combined with the result for $I_{SC}$ = 0.673 µA and $V_{OC}$ = 436 mV, $\eta$ was found to be 2.43 × 10$^{-5}$% and *FF* was 0.248.

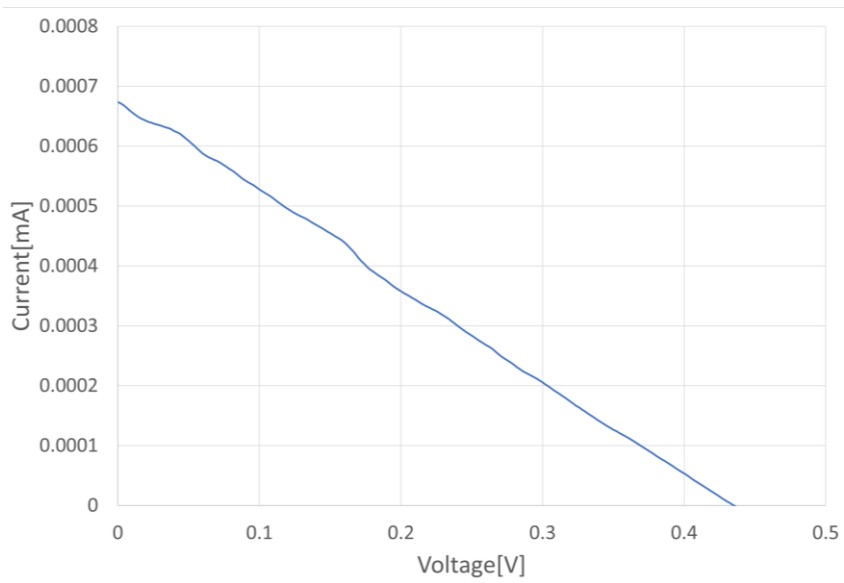

**Figure 11.** Current–voltage characteristics of the sample.

The proposed paper DSSC using gel electrolytes has shown potential in improving *FF* compared to our previous paper DSSCs [15] that use liquid electrolytes. However, $\eta$ of our paper DSSC using gel electrolytes was found to be lower. It is known that the improvement of the internal resistance is crucial to enhance *FF*. In the paper DSSC using liquid electrolytes reported in a previous study, the electrolyte penetrated into various parts of the sample, causing paper swelling and other problems that affected the electrical network of CNTs, resulting in higher internal resistance. The adoption of gel electrolytes in this study could have helped mitigate this negative influence, leading to an improvement in *FF*.

It was reported that paper DSSCs with liquid electrolytes operated for only about 20 min due to evaporation and leakage of the electrolyte. In contrast, by introducing the gel electrolytes, it was confirmed in this study that the working time could be extended to more than 110 min, as shown in Figure 12. Although the conversion efficiency of the sample with the gel electrolytes was lower than the previous one, this result of improved lifetime is very important in view of practical applications.

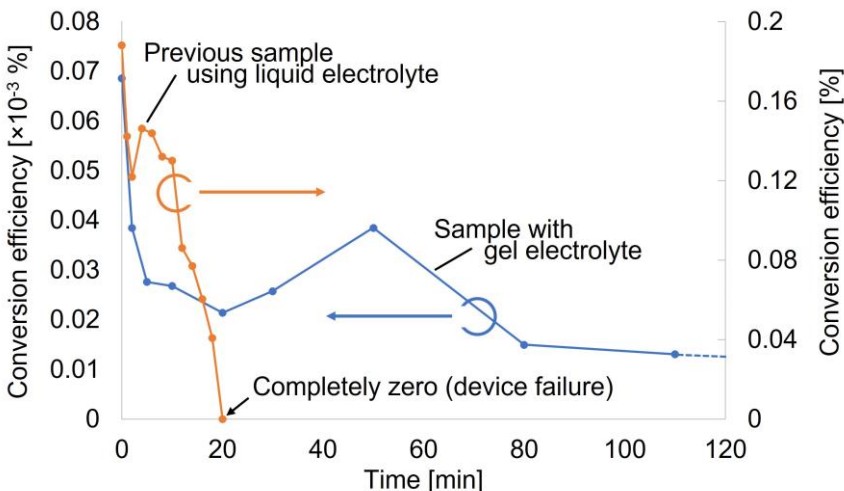

**Figure 12.** Comparison of operating time between previous and proposed samples.

*Modification of Paper DSSC to Improve Its Performance*

To improve the performance of our paper DSSC, we here focus on the inner resistance of the metallic CNT composite papers and the composition of the gel electrolyte. To reduce the inner resistance of the metallic CNT composite papers, we change the used CNT for the composite paper. When preparing the CNT composite paper, the properties of the used CNT influence the overall performance of the composite paper. In general DSSCs, it is known that reducing the resistance of electrodes leads to a higher internal current, resulting in an increased output current and an improved conversion efficiency. Therefore, here we checked whether it is possible to reduce the internal resistance of the composite paper by changing the metallic CNTs used. Table 3 shows the parameters of SGCNT and HiPco (NanoIntegris Inc., Boisbriand, Quebec, Canada) that we chose here because of a long well-known CNT, and the parameters of the CNT composite paper based on each CNT. Figure 13 shows the produced composite papers.

**Table 3.** Parameters of HiPco and SGCNT, and composite papers based on each CNT.

| | Diameter [nm] | Length [μm] | Volume Resistivity of Composite Paper [$\times 10^3$ Ω·m] |
|---|---|---|---|
| SGCNT | around 4.0 | around 400 | 25–75 |
| HiPco | around 1.0 | around 0.5 | 148–185 |

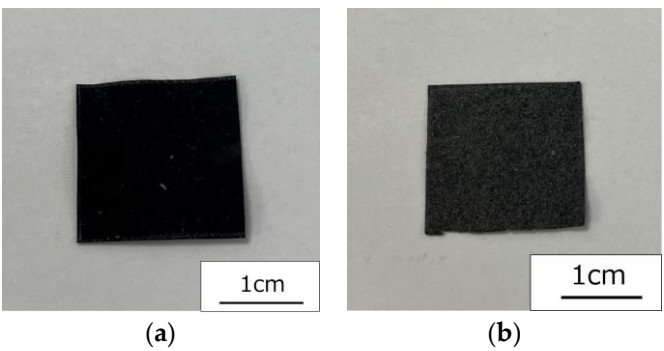

**Figure 13.** Produced CNT composite papers. (**a**) SGCNT-based composite paper and (**b**) HiPco-based composite paper.

As a result, the composite paper with HiPco CNTs did not result in a reduction in resistance compared to that with SGCNTs. This is thought to be mainly due to differences

in the length of the CNTs and the CNT network formed in the composite paper. However, we were also able to discover the advantages of using HiPco CNTs. When we fabricated our paper DSSC using the HiPco-based metallic CNT composite paper and the semiconducting CNT composite paper described above, and evaluated its conversion efficiency and *FF*, we confirmed the improvement in the conversion efficiency and *FF* compared to those using SGCNTs. Specifically, by measurement and calculation, the conversion efficiency was found to be $7.48 \times 10^{-5}\%$, and *FF* was found to be 0.276. This is considered to be due to the larger surface area in contact with the electrolyte because of the narrower diameter of the HiPco CNTs. It is considered that the enlarged surface area allows more charge to be handled efficiently.

Based on this result, we considered the use of HiPco-based composite paper to be effective, and tested changes in the composition of the gel electrolyte for this. In this study, the iodine-to-potassium-iodide ratio was set at about 19.1 to 100 to ensure complete reaction as described in Section 2.2. Here, the composition of the gel electrolyte was changed by doubling the amount of PEG and PVDF used, keeping the ratio fixed, and its influence on paper DSSC was evaluated.

As a result, the photovoltaic conversion efficiency was improved to be $2.02 \times 10^{-3}\%$ with *FF* of 0.164. Figure 14 shows the *I-V* characteristics of the sample. Although *FF* was slightly lower, an increase in maximum power was obtained, and overall the conversion efficiency was improved. This result suggests that increasing PEG and PVDF may contribute to the improvement of the conversion efficiency. There is a report by another research group that shows that changes in the amount of PEO (similar chemical of PEG) can change the flow of ions [23], and the results obtained in this study are consistent with this reported phenomenon. The previous study [24] indicated that similar changes were observed when PVDF was added to the PEO electrolyte. These findings are consistent with the results of this study. The paper also mentioned that the introduction of PVDF, which contains highly electronegative fluorine, into the PEO electrolyte system has a significant impact on the interfacial electron transfer from the photoanode to tri-iodide in dye-sensitized nanocrystalline solar cells. Furthermore, the paper suggested an enhancement in the photocurrent density, and if similar effects are obtained in this study, it could lead to improved power generation efficiency.

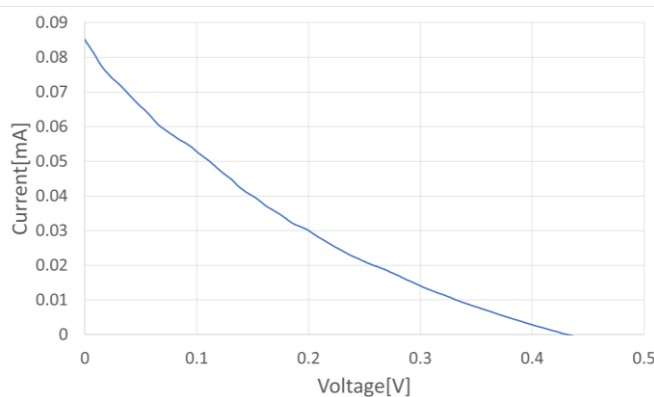

**Figure 14.** *I-V* characteristics of modified paper DSSCs.

However, the *FF* was lower than that of the base sample as described above, suggesting that the composition of the gel electrolyte is important for both conversion efficiency and *FF* improvement. However, some other samples prepared in the same way were found to have both improved conversion efficiency and *FF*, although imperfectly. We will clarify the cause of the difference in future works and develop paper DSSCs with better performance.

## 4. Conclusions

In this study, we proposed the use of gel electrolyte for a paper DSSC based on our CNT composite papers. Our CNT composite paper inherits both the metallic and semiconducting properties of the contained CNTs. This enables the CNT composite paper to be used as positive and negative electrodes for the DSSCs, and our paper DSSCs were able to be prepared. Compared with the general DSSCs, the proposed paper DSSC may have some inferior points. However, we believe it has the following advantages. For example, CNTs are considered one of the most promising materials to replace platinum electrodes used in general DSSCs. Our CNT composite paper also has advantages such as eco-friendliness, low cost, and ease of fabrication. It has also been experimentally demonstrated as a possibility to be used as an electrode in DSSCs [15]. Furthermore, there is the possibility of realizing things that are not possible with ordinary DSSCs, such as the introduction of a three-dimensional structure based on an "origami" technique, which takes advantage of the fact that it is paper and can be easily made into large-area products. Therefore, we believe that through further optimization of the cell, there is still great potential for the development of CNT composite paper-based DSSCs.

As a first step of this study, the application of gel electrolytes in paper DSSCs was investigated in order to improve their stability. Specifically, conventional potassium iodide liquid electrolytes for DSSCs were gelated and the produced gel electrolyte was applied to paper DSSCs using PEG and PVDF as polymers. In the preparation of paper DSSCs, the ungelatinized electrolyte was firstly obtained by evaporating most of the contained acetonitrile solution and subsequently, 400 μL of the ungelatinized gel electrolyte was dropped between the metallic and the semiconducting CNT composite papers to gel it on the composite paper.

The use of gel electrolytes showed potential in improving *FF* compared to previous paper DSSCs [15] that use liquid electrolytes. By using a gel electrolyte, the *FF* of the paper-DSSC was increased from 0.12 to 0.248, and the lifetime was increased from 20 min to over 110 min, the electrolyte leakage and evaporation problems of the paper DSSCs were solved to some extent. However, in this step, the conversion efficiency of our paper-DSSCs using gel electrolytes was found to be lower.

On this basis, modifications were made to the metallic CNT composite paper and the gel electrolyte to improve the conversion efficiency of our paper DSSC. As a result, *FF* was 0.164, and the conversion efficiency was improved from $2.43 \times 10^{-5}\%$ to $2.02 \times 10^{-3}\%$. Several factors may be responsible for the low conversion efficiency of the paper DSSC. In a previous study [15], a photovoltaic conversion efficiency of 0.18% was reported under liquid electrolyte conditions using CNT composite papers attached to specially shaped electrodes that efficiently collect excited electrons. Based on the results obtained in the previous studies and this study, the main reasons for the low conversion efficiency are considered to be as follows: (1) high internal resistance, structural instability, and uninventive electrode shape; (2) composition of the gel electrolyte is not optimal; (3) dye fixation on CNTs is not optimal; and (4) the relationship between the work function of the CNT used (P-type semiconducting type) and the dye used is uncertain. As mentioned above, there are certainly some difficult issues that remain. However, as proposed in this paper, the possibility that the use of gel electrolytes can dramatically improve the problem of short lifetime, which has been an issue until now, is clarified. We believe that this paper DSSC will be used in our daily life in the near future.

**Author Contributions:** All authors contributed equally. Conceptualization, Y.K. and T.O.; methodology, Y.K. and T.O.; validation, Y.K. and T.O.; formal analysis, Y.K.; investigation, Y.K. and T.O.; resources, Y.K. and T.O.; data curation, Y.K.; writing—original draft preparation, Y.K.; writing—review and editing, Y.K.; visualization, Y.K. and T.O.; supervision, Y.K.; project administration, Y.K. All authors have read and agreed to the published version of the manuscript.

**Funding:** This research received no external funding.

**Data Availability Statement:** The data described in the manuscript are available from all authors on reasonable request.

**Acknowledgments:** We are grateful to all other contributors for providing us with the early data presented at the 70th JSAP Spring Meeting held in Tokyo, Japan on 16 March 2023, and to Masahiro Yano and Koya Arai of the Innovation Center at Mitsubishi Materials Corporation, Japan for their valuable comments and support.

**Conflicts of Interest:** The authors declare no conflict of interest.

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
