# Peer review of "Unique Dye-Sensitized Solar Cell Using Carbon Nanotube Composite Papers with Gel Electrolyte"

_jcs, doi:10.3390/jcs7060232_

Round 1
Reviewer 1 Report (Previous Reviewer 3)
Only few minor spelling/syntax errors
Page 2, Line 62: 'semiconducting or metallic property' instead of 'semiconducting and metallic property'
Page 7, Line 199: Isc instead of Ioc
Page 9, Line 253: 'resulting in an increased output current and an improved conversion efficiency' instead of 'resulting in increased output current and improved the 253 conversion efficiency'
No further comments
Author Response
Dear Reviewer,
We would like to express our appreciation for your review of our paper. We are glad to hear that you recommend the paper for publication pending minor revisions. We value your feedback and suggestions for improvement.
We have corrected what you have pointed out.
We thank you for your time and expertise, and we look forward to your final decision on the publication of our paper.
Thank you once again for your valuable contribution to our research.
Sincerely,
Yi Kou. (Corresponding author)
Yokohama National University
79-5, Tokiwadai, Hodogaya-ku, Yokokohama, 240-8501, Japan
+81-45-339-4125
kou-yi-zt@ynu.jp
Reviewer 2 Report (Previous Reviewer 2)
The authors have made the necessary revision and now it is ready for publication.
English is OK
Author Response
Dear Reviewer,
We would like to express our gratitude for reviewing our paper and for your positive assessment. We greatly appreciate your support and feedback.
We will work diligently to submit the final version to you as soon as possible. We thank you once again for your time and consideration, and we look forward to your final decision regarding the publication of our paper.
Thank you once again for your valuable contribution to our research.
Sincerely,
Yi Kou. (Corresponding author)
Yokohama National University
79-5, Tokiwadai, Hodogaya-ku, Yokokohama, 240-8501, Japan
+81-45-339-4125
kou-yi-zt@ynu.jp
Reviewer 3 Report (Previous Reviewer 1)
The authors have addressed the all questions.
Author Response
Dear Reviewer,
We would like to express our gratitude for reviewing our paper and for your positive assessment. We greatly appreciate your support and feedback.
We will work diligently to submit the final version to you as soon as possible. We thank you once again for your time and consideration, and we look forward to your final decision regarding the publication of our paper.
Thank you once again for your valuable contribution to our research.
Sincerely,
Yi Kou. (Corresponding author)
Yokohama National University
79-5, Tokiwadai, Hodogaya-ku, Yokokohama, 240-8501, Japan
+81-45-339-4125
kou-yi-zt@ynu.jp
Round 2
Reviewer 1 Report (Previous Reviewer 3)
No further comments
This manuscript is a resubmission of an earlier submission. The following is a list of the peer review reports and author responses from that submission.
Round 1
Reviewer 1 Report
The authors reported that the “Unique dye-sensitized solar cell using carbon nanotube composite-papers with gel electrolyte”. They prepared a unique dye-sensitized solar cells (DSSCs), paper-DSSCs based on carbon nanotube composite papers, and the use of a gel electrolyte for the paper DSSC. The polymer gel electrolyte was synthesized using a copolymer of polyethylene glycol and polyvinylidene fluoride as a matrix, mixed with iodine and potassium iodide. The resulting our paper DSSC had a fill factor of 0.248 and a conversion efficiency of 2.43 × 10-5 % with an extended working time (lifetime) of more than 110 min. after the evaluation. I think it could be published after the minor revision. The details were as follows.
1. Is the dye molecule used in the article synthesized by the author? If it was synthesized by oneself, please provide relevant characterization results. If not, please indicate the source of the drug sample.
2. It is difficult to reflect the role of dye molecules in Figure 1. It is recommended that the author make appropriate modifications to the schematic diagram.
3. TiO2 is very important in many fields, and regarding its applications, the author needs to refer to this literature to make it more abundant. (ACS NANO, 2022, 16, 4487)
There is no comment in this section.
Reviewer 2 Report
Compared to its counterpart peroveskite solar cell, the progress of DSSCs is not made so fast. In this work, the authors tried to improve a DSSCs based on carbon-nanotube (CNT) composite papers and a gel electrolyte was used instead of liquid-type electrolyte in order to increase safety and durability. The authors then used a copolymer of polyethylene glycol (PEG) and polyvinylidene fluoride (PVDF) as a matrix, mixed with iodine and potassium iodide to synthesize the polymer gel electrolyte. They achieved the paper-DSSC with a fill factor of 0.248 and a conversion efficiency of 2.43 × 10-5 % with an extended working time of more than 110 min. Further they utilized the metallic CNT composite paper and the gel electrolyte in DSSCs, resulting in increased a conversion efficiency of 2.02 × 10-3 %. There are some issues need to be clarified before the acceptance of the manu for publication:
(1) Apparently, the conversion efficiency of the proposed DSSCs is bad. It is not suggested to publish this paper before its performance has been improved greatly. What factors may be the major reasons for the low conversion efficiency?
(2) The technological route to use CNT composite paper may not be suitable for DSSCs.
(3) In line 283, “This result suggests that increasing PEG and PVDF may contribute to the improvement of the conversion efficiency.” Please list two sets of data for comparison and explanation.
(4) In section 3.2, the authors wanted to reduce the internal resistance of the composite paper by changing the composition of the metal carbon nanotubes, why only SCGNT and HiPco were considered and not chirality CNTs?
The authors can express their opinion well in English
Reviewer 3 Report
The role of the nanostructured TiO2 layer at the anode of a dye sensitized solar cell is to provide a high internal surface area for the adsorption of the dye molecules. The TiO2, due to its high band gap, absorbs light below 400 nm. Therefore, all the incident light is absorbed by the dye molecules. The above is a basic principle for DSSCs.
In the current study, the major part of the incident light is absorbed by the CNT composite paper (the anode). Moreover, there is no indication of an efficient sensitization, since the CNT paper after sensitization has a yellowish appearance and not a purple one. Propably, these are the two reasons that could explain the extremely low efficiency. Finally, there is no indication regarding the reproducibility of the reported efficiency values.
In conclusion, in my opinion, even though the idea of a paper-based DSSC is interesting, the inappropriate anode material, due to its absorption, the extremely low efficiency and lack of reproducibility, indicate that the current study cannot be published
No comments
